# Uncovering the Role of the Early Visual Cortex in Visual Mental Imagery

Nadine Dijkstra

Department of Imaging Neuroscience, Institute of Neurology, University College London, London WC1E 6BT, UK; n.dijkstra@ucl.ac.uk

**Abstract:** The question of whether the early visual cortex (EVC) is involved in visual mental imagery remains a topic of debate. In this paper, I propose that the inconsistency in findings can be explained by the unique challenges associated with investigating EVC activity during imagery. During perception, the EVC processes low-level features, which means that activity is highly sensitive to variation in visual details. If the EVC has the same role during visual mental imagery, any change in the visual details of the mental image would lead to corresponding changes in EVC activity. Within this context, the question should not be whether the EVC is 'active' during imagery but how its activity relates to specific imagery properties. Studies using methods that are sensitive to variation in low-level features reveal that imagery can recruit the EVC in similar ways as perception. However, not all mental images contain a high level of visual details. Therefore, I end by considering a more nuanced view, which states that imagery can recruit the EVC, but that does not mean that it always does so.

**Keywords:** mental imagery; visual perception; early visual cortex





## 1. Introduction

Close your eyes and imagine an apple. Is this experience similar to actually seeing an apple or is it more like thinking about the idea of an apple? Answering this question, whether imagery relies on depictive or symbolic representations, has been referred to as the 'imagery-debate'. This debate has dominated mental imagery research during the end of the last century and the beginning of this one [1,2]. The development of cognitive neuroimaging promised a way to settle the debate once and for all: if mental images, like perceived images, are represented in the retinotopically organized early visual cortex (EVC), this would be knock-out evidence in favor of a depictive view. However, at this moment, more than three decades after the first neuroimaging studies on mental imagery, the role of the early visual cortex during visual mental imagery remains a topic of intense debate [3–6].

In this paper, I argue that in order to make progress, we need to move away from the question of *whether* the EVC is involved in mental imagery and instead move towards elucidating exactly *how* it is involved. I argue that many of the discrepancies in the literature can be explained by the unique methodological and conceptual challenges associated with the functional role of the EVC. I will start by explaining that the question of whether the EVC is 'active' or not during imagery is conceptually nonsensical in light of what we know about the functionality of the EVC. Instead, to elucidate the involvement of the EVC during mental imagery, its unique organization needs to be considered both in the experimental design as well as in the data analysis. I will discuss examples of studies that do this and explain what they tell us about how the EVC is involved in visual mental imagery. I focus on the volitional use of mental imagery and will mostly discuss studies in which participants are explicitly instructed to form mental images. In Section 4, I will further discuss to what extent these conclusions generalize to more implicit forms of imagery.

Taken together, I propose that the current evidence suggests that the EVC represents fine-grained visual details during mental imagery in the same way as it does during

perception. I will end by discussing how this account can explain contradictory evidence and highlight directions for future research.

## 2. Organizational Principles of the Early Visual Cortex

Decades of neurophysiological research have led to a highly detailed characterization of how the early visual cortex responds to external visual signals. Activity in the EVC is retinotopically organized such that things that are spatially close together in the outside world elicit activity in neurons that are close together on the cortical sheet of the EVC. Furthermore, neurons in the EVC have small receptive fields so that each neuron only responds to signals coming from a very small portion of the visual field. Together, this means that activity in the EVC is highly sensitive to changes in low-level features of visual signals, such as their exact distribution of edges, location and orientation [7]. This is in contrast to neurons in more high-level visual areas that have much larger receptive fields and are therefore able to represent more 'high-level' features, such as the semantic meaning of a stimulus, irrespective of the specific low-level instantiation of that stimulus [8]. It is important to note that this distinction between low- and high-level sensitivity is not discrete. Specifically, activity in the EVC has also been shown to be modulated by high-level properties such as semantic scene context [9,10], and activity in high-level cortex has also been shown to be influenced by certain low-level features such as location [11]. However, in general, activity in the EVC is much more sensitive to variation in low-level features [12]. This means that, for example, the same neurons in the high-level visual cortex might fire in response to a chair from different viewpoints, whereas the pattern of activity in the EVC will be widely different in each specific instance.

The properties of the neurons in the EVC have the consequence that any change in low-level features of visual signals, for example, the exact orientation of a stimulus, will lead to a corresponding change in EVC activity (Figure 1A). While looking outside your window at a tree, any displacement of the branches or the leaves due to the wind will result in a change in activity in your early visual cortex. Furthermore, any movement of your eyes will change in which receptive fields the visual signals of the tree fall, again leading to corresponding changes in EVC activity. If you were to now test whether any given neuron in the EVC was consistently active when you were looking at this tree, you would likely find very few neurons that showed consistent activity. Based on this, you could wrongly conclude that the EVC was not involved in the perception of the tree. This is why most visual neuroscience experiments studying the EVC are done with controlled repetitions of the exact same set of simple stimuli, presented in exactly the same location on the screen, with the participant's head firmly fixated on a chin rest and taking dedicated measures to minimize eye-movements as much as possible. Under these controlled conditions, it is much more likely to observe consistent activity in EVC neurons.

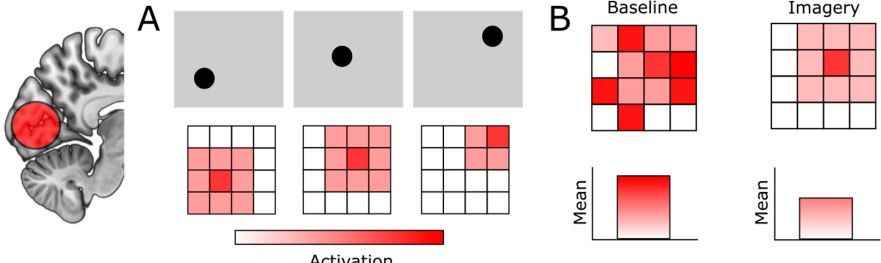

**Figure 1.** Challenging properties of early visual cortex (EVC) activity during imagery. (**A**) Activity in the EVC is highly sensitive to changes in low-level features, such as where in visual space a stimulus is located. Top row indicates toy visual signals, spreading over the visual field. Bottom row indicates the activity pattern in a grid of EVC neurons. (**B**) Imagery is thought to be instantiated through inhibitory feedback connections, leading to the sharpening of representations rather than increases in general activation levels. This means that the EVC can still be involved in imagery even if there is no change or even a decrease in general activation.

### 3. The Challenges of Studying the Early Visual Cortex during Mental Imagery

Now, consider imagining a tree in your mind's eye. How stable are the low-level features of this mental image? Are the locations of the branches fixed and clear from one moment to the next? Are you able to hold a clear and stable image without moving your eyes? A recent study showed that when asked to freely imagine a scene, most participants leave out many low-level details [13]. Furthermore, some people might even experience mental images in a kind of separate space that is decoupled from the physical environment [14]. Herein lies the main difficulty of studying the role of the EVC in imagery: Given the sensitivity of activity in the EVC to changes in low-level features, the fleeting and undefined nature of mental images makes it very hard to determine any consistent relationship. In order to properly study the role of the EVC in imagery, it is therefore essential to have a firm handle on the low-level features of the mental images under investigation.

One way to control the low-level features of mental images is by instructing participants to imagine very simple stimuli. One of the earliest studies using this approach was by Klein and colleagues in 2004 [15]. In this study, participants were instructed to view and imagine horizontally and vertically oriented bow-tie stimuli while their brain activity was measured using functional magnetic resonance imaging (fMRI). Crucially, the exact spatial location and orientation of the to-be-imagined stimuli were clearly defined. The results revealed that imagining a horizontal stimulus activated the horizontal meridian within the EVC while imagining a vertical stimulus activated the vertical meridian, in line with retinotopic involvement of the EVC during imagery [15]. Similar approaches have been used to show retinotopic encoding in the EVC of imagined domino patterns [16] and letters [17].

Another unique challenge of studying the EVC during imagery is that the way that EVC activity is instantiated during imagery is different from perception. Specifically, during imagery, activity in the EVC is assumed to be instantiated via feedback connections from high-level areas [18–20]. Importantly, in contrast to feedforward connections, feedback connections tend to predominantly *modulate* neural activity, changing existing firing rates via gain control, but usually without driving neurons to fire action potentials directly [21,22]. In line with this, it has recently been proposed that imagery inhibits activity in irrelevant neuronal populations rather than directly increasing activity in the populations representing the imagined stimulus [23]. This is similar to the 'sharpening' process assumed to underlie attention [24] and expectation [25]. Instead of modulating bottom-up signals like during attention and expectation, imagery is assumed to modulate spontaneous, baseline fluctuations in brain activity, carving out stimulus representations by inhibiting irrelevant activity [5,26]. Importantly, if imagery is instantiated through inhibition of irrelevant activity, overall activity levels in the EVC might not increase and might even decrease compared to baseline (Figure 1B). This could explain the observation that aphantasia—the absence of visual mental imagery—has been associated with hyperactivity of the EVC [27,28]. Importantly, however, if the EVC is involved in imagery, the relative activity pattern should still be informative of which stimulus participants were imagining, even if general activation levels do not increase.

Testing whether neural activity patterns contain information about perceived or imagined stimuli is the main goal of 'decoding' techniques in cognitive neuroscience. In this context, decoding refers to the use of machine learning algorithms to describe how neural activity patterns relate to different stimuli or conditions [29]. In a standard imagery decoding experiment, a decoder is first 'trained' to capture the variation in brain activity patterns during the perception of different stimuli. This perception-decoder is then applied to the brain activity during imagery of the same stimuli, giving a 'guess' of which stimulus the participant was imagining. If the perception-classifier is able to accurately guess or 'decode' the imagined stimulus, it can be concluded that the pattern of neural activity during imagery is similar to that during perception in that brain area. Using this technique, significant cross-decoding between imagery and perception in the EVC has been shown for gratings, letters, objects and shapes [17,30–32].

Furthermore, to more directly test whether low-level visual features are encoded in EVC activity patterns, researchers have used the so-called feature encoding models [33]. These are models based on computer vision algorithms that describe complex images in terms of unique combinations of simple, low-level features. These models can, in turn, be combined with models that describe activity in different brain areas in response to the same low-level features to predict what the brain activity would be like for an entirely new set of complex images [34]. For example, one study used a low-level feature encoding model to first determine how images of complex artworks could be described in terms of a combination of Gabor wavelets [35]. Next, the activity in the EVC during the perception of the same Gabor wavelets was measured to create a corresponding low-level feature model of EVC activity. Applying the transformation between Gabor wavelets and the artworks to EVC activity during the imagery of different artworks led to the successful prediction of which artwork participants were imagining [35]. Several other recent studies have used similar computer vision approaches to reveal low-level feature encoding of visual mental imagery in brain activity patterns [36,37].

Together, these studies highlight that in order to capture EVC involvement during visual mental imagery, it is essential to use the right experimental and analytical approach. In this context, investigating whether the EVC is 'active' or not without specifying the exact way in which its activity relates to specific properties of the mental image is unlikely to reveal consistent involvement. For example, consider an experiment in which, on each trial, participants are asked to imagine a specific stimulus but are not asked to imagine that stimulus in a specific size or at a specific location. Due to the organization of the EVC, each of those imagery instances would be associated with different patterns of activity. Averaging over trials might, therefore, lead to the incorrect conclusion that imagery did not recruit the EVC. This can also explain why meta-analyses investigating the activation of imagery of different stimuli over different studies are ill-placed to find activation in the EVC and are instead more likely to find consistent involvement of higher-level visual regions [6].

## 4. The Role of the Early Visual Cortex in Visual Mental Imagery

Considering the evidence until now, I propose that the EVC represents fine-grained visual details during imagery in similar ways as it does during perception, in line with a depictive view. However, importantly, that does not mean that all instances of visual mental imagery will activate the EVC. Consider, for example, a recent study in which participants were asked to imagine a person walking into a room and knocking a ball off a table [13]. Afterwards, participants were asked several questions regarding the visual details of their mental imagery, i.e., 'did you imagine the color of the ball', 'the clothes the person was wearing', etc. It turned out that most participants did not imagine the majority of basic visual details. The authors concluded that 'while imagination may indeed be a good artist, it's on a deadline, and stingy about paint' ([13], p. 20). In other words, imagination *can* create vivid, detailed scenes, but that requires effort, so if it is not necessary, those details will be omitted.

This means that EVC activity is likely not always observed during mental imagery, but mostly when low-level details are indeed imagined. The idea that EVC activity depends on whether the imagery in question requires fine-grained details is in line with early observations that noted that the EVC is more likely to be activated in imagery tasks that require high-resolution discrimination judgments [38]. Furthermore, the observation that the vividness of mental imagery correlates with perception-imagery cross-decoding accuracy in EVC [31,39,40] is also in line with the idea that more detailed mental images are more likely to recruit the EVC.

One line of research that has been taken as evidence against the involvement of the EVC in visual mental imagery comes from neuropsychological observations of intact visual mental imagery after lesions to the EVC [3,41]. One way to align these findings with the previously mentioned neuroimaging research is with the fact that imagery does not

always contain fine-grained visual details [42]. This would mean that patients without a functioning EVC might still be able to generate mental images, but that these mental images would not contain the fine-grained visual details that require a functioning EVC. Future studies using controlled psychophysical paradigms in these patients are required to determine whether this is indeed the case.

Another potential argument against the involvement of the EVC in imagery comes from recent studies that show that imagery can be decoded from EVC activity in aphantasia [43–45]. However, the claim that EVC representations are necessary to imagine low-level features does not mean that these representations are also sufficient for conscious visual imagery. One possibility is that these low-level EVC representations need to be coupled with higher-level representations in frontoparietal areas in order to reach awareness, as proposed by higher-order theories of consciousness [46–48]. In line with this, a recent study found that there were no differences in visual regions between aphantasia and controls but that in aphantasia, visual regions were functionally disconnected from the fronto-parietal cortex [49].

The EVC also seems to represent internally generated visual information in contexts where imagery is not explicitly instructed. For example, EVC activity tracks the generated stimulus during mental rotation [30,50] as well as during visual working memory [30,51], even though both of these processes have been shown to be largely unaffected in aphantasia [52,53], suggesting that they might not require conscious mental imagery. These findings suggest that the EVC represents internally generated visual information irrespective of whether these signals are instantiated through volitional, conscious imagery or not.

## 5. Conclusions and Future Directions

The question of whether visual mental imagery recruits the early visual cortex has been at the core of the imagery-debate for decades. Initial neuroimaging findings were inconclusive, with some studies finding reliable activity while other studies did not. In this paper, I have suggested that the inconsistency in the literature can be explained by the unique challenges associated with studying the EVC during visual mental imagery. The EVC is known for representing fine-grained visual details during perception. However, compared to perception, imagery tends to be associated with weaker and less consistent visual signals. Moreover, some instances of mental imagery might contain very few fine-grained visual details at all. These challenges require specific experimental and analytical approaches, such as using very simple stimuli or quantifying expected activity patterns with sophisticated models. The results of studies using these approaches consistently show that the EVC represents low-level visual details during mental imagery in similar ways as it does during perception.

A number of questions regarding the exact role of the EVC in visual mental imagery remain open. Firstly, the exact relationship between EVC representations and conscious imagery experience remains unclear. Several studies have shown a relationship between EVC involvement and subjective reports of imagery vividness [31,39,40]. However, recent studies suggest that EVC representations can be present in the complete absence of conscious visual imagery [43–45]. Future research is necessary to investigate which factors determine whether imagery-related EVC activity becomes conscious [54].

Relatedly, exactly how visual mental imagery representations are instantiated in the EVC remains unclear. The current consensus is that this happens through (inhibitory) feedback connections from high-level areas [4,5]. However, it is not clear exactly which high-level regions are referred to and exactly how activation flows through these different regions. Several studies have found increases in coupling between the dorsolateral prefrontal and visual cortex during imagery [19,20]. Other studies have shown increases in coupling between parietal and visual regions [18]. Lastly, research on scene imagery has focused on the role of medial frontal and hippocampal memory systems in generating mental images [55]. Further research is necessary to fully elucidate the neural network involved in generating EVC activity during mental imagery.

Finally, given that imagery seems to be able to recruit the EVC in a similar way to perception, one important open question is how the brain keeps track of which visual signals reflect external reality and which reflect internal imagination. One hypothesis has been that the distinction is made based on whether the middle layer of the EVC is involved, which is where bottom-up sensory signals arrive in the cortex [56]. Another hypothesis is that the feeling of 'realness' is determined by whether or not the superficial layers of the EVC are activated [57]. Initial results show that during mental imagery, only deep layers of the EVC are involved. In contrast, superficial layers do become activated during seemingly 'real' illusory perception [57]. Another not mutually exclusive hypothesis is that whether EVC signals are considered real or not depends on whether they are strong enough to cross a 'reality threshold' [40]. Finally, other proposals have suggested that external, feedforward and internal feedback signals might be separated in time, where feedforward signals might be carried by faster oscillations in the gamma band, while feedback signals are carried by slower oscillations in the alpha/beta band [58–62]. Further elucidating how the brain keeps apart EVC signals of imagery and perception has important implications for our understanding of when this line breaks down in hallucinations.

To conclude, developments in both experimental design and analytical approaches are converging on the idea that the EVC is recruited when low-level visual details are imagined. Future research is needed to further characterize exactly how activity in the EVC is instantiated during imagery and how this activity leads to the unique conscious experience characteristic of our imagination. Answering these remaining questions will further increase our fundamental understanding of the neural correlates of consciousness as well as provide novel perspectives on understanding disorders of mental imagery.

**Funding:** This research was funded by a Marie Curie grant from the European Union Horizon 2020 program [882832/MSCA/IF/EF/ST], UK Research and Innovation (UKRI) under the UK government's Horizon Europe funding guarantee [selected as ERC Consolidator, grant number 101043666] and an Adversarial Collaboration Grant from the Templeton World Charity Foundation [TWCF number 22032].

**Acknowledgments:** The author would also like to thank Benjy Barnett for his comments on an earlier version of this manuscript.

**Conflicts of Interest:** The author declares no conflict of interest.

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
