# Peer review of "Uncovering the Role of the Early Visual Cortex in Visual Mental Imagery"

_2411-5150_

Round 1
Reviewer 1 Report
Comments and Suggestions for Authors
The author addressed a specific challenging issue that concerns the role of early visual cortical processing in mental imagery.
The review is convincing, providing a clear and informative analysis of both the experimental and conceptual problems to disentangle the role depictive and symbolic representations in visual mental imagery.
Author Response
I would like to thank the reviewer for their positive evaluation of the manuscript.
Reviewer 2 Report
Comments and Suggestions for Authors
I found the manuscript very interesting. The literature discussed is recent, arguments are robust, and the perspective is agreeable. I commend the author for the clarity of presentation. However, I have four points needing integration:
1) Regarding imagery in the manuscript, does it exclusively refer to the intentional and active creation of mental images, or does it also encompass cases where the task doesn't explicitly require mental image creation but involves manipulating visual information (mental rotation and similia)? Has the author selected for the review solely literature about voluntary imagery tasks, or have both cases been considered? A clearer operational definition of what the author mean with imagery in this manuscript would be beneficial to the introduction. My fourth point (below) assumes the inclusion of tasks not explicitly related to imagery but where imagery is present.
2) From line 108 to 125, especially from 115 onwards when discussing how imagery inhibits activity in irrelevant neuronal populations, I suggest providing more clarification. When referring to modulating activity, does it include spontaneous and/or sub-threshold pre-mental imaging activity? If so, how is relevant activity distinguished from irrelevant activity? And why does inhibiting irrelevant activity promote the emergence of mental images? The 'sharpening' process linked to attention (Bartsch et al., 2017) and expectation (Kok et al., 2012) modulates feedforward (suprathreshold stimulus-related activation). What modulates the feedback of imagery? Clarifying this point is essential.
3) In the same section discussing the involvement of the visual cortex in imagery, what is the relationship between feedback (imagery) and retinal feedforward? Are they distinct neural populations in the early visual cortex, or are they the same population? Is there competition for resources between the two? Could there be temporal alternation? These questions are relevant and should be at least partially addressed in the text. The following excerpt illustrates my point, but the author may skip it if deemed irrelevant: studies on mice show that the primary visual cortex is involved in both feedforward and feedback processes at distinct time intervals (Keller et al., 2020). Wilming et al. (2020) found a dissociation between endogenous (feedback) and exogenous (feedforward) components in the early visual cortex (including V1) in terms of frequency bands. Gamma conveyed stimulus-dependent information, while alpha carried endogenous information related to perceptual choice. These findings suggest a temporal alternation between feedback and feedforward processes, possibly involving overlapping neural populations but at different times (Semedo et al., 2022, Contemori et al., 2022).
4) In the introduction, the early visual cortex is described as a retinotopic structure. Please note that the foveal cortex has been shown to be involved in non-retinotopic activity (i.e., Foveal feedback, Williams et al., 2008, but see Oletto et al., 2022, for a review). This non-retinotopic feedback process appears to be related to some form of imagery process involving mental manipulation of the stimulus, as its timing can be modulated by a mental rotation task (Fan, 2016). The author mentions at line 169 and onwards that the EVC represents fine-grained visual details during imagery, and without vivid details, it is not involved in imagery. This claim aligns with studies on non-retinotopic Foveal Feedback. For instance, Fan (2016) demonstrates that the feedback does not activate for blurred objects. Additionally, Contemori (2023) shows that interfering with feedback not only affects task sensitivity but also introduces perceptual biases in decision criteria. The line of research on Foveal Feedback seems closely connected to what is described in the review, and there are interesting links that could provide relevant insights into the topic.
Comments on the Quality of English Language
I commend the author for the clarity of presentation.
Author Response
Reviewer 2: I found the manuscript very interesting. The literature discussed is recent, arguments are robust, and the perspective is agreeable. I commend the author for the clarity of presentation. However, I have four points needing integration.
Response: I would like to thank the reviewer for their overall positive evaluation and helpful comments. I address each of them in detail below.
Comment 1: Regarding imagery in the manuscript, does it exclusively refer to the intentional and active creation of mental images, or does it also encompass cases where the task doesn't explicitly require mental image creation but involves manipulating visual information (mental rotation and similia)? Has the author selected for the review solely literature about voluntary imagery tasks, or have both cases been considered? A clearer operational definition of what the author mean with imagery in this manuscript would be beneficial to the introduction. My fourth point (below) assumes the inclusion of tasks not explicitly related to imagery but where imagery is present.
Response: I agree with the reviewer that it is important to have a clearer operational definition of imagery in the paper. The imagery studies that I discuss mostly pertain to explicit, volitional imagery, except the Cabbai preprint where they directly compared voluntary and involuntary imagery, however, I would expect that they also hold for tasks that require manipulating visual information, even when imagery is not explicitly instructed. I have added this to the manuscript as follows:
In the introduction on page 1, line 41: “…they tell us about how the EVC is involved in visual mental imagery. I will focus on studies in which participants are explicitly instructed to form mental images but propose that the conclusions generalize to more implicit forms of imagery, which I will discuss in more detail in section 4.”
In section 4, page 5, line 241: “The EVC also seems to represent internally generated visual information in contexts where imagery is not explicitly instructed. For example, EVC activity tracks the generated stimulus during mental rotation (Albers et al., 2013; Iamshchinina et al., 2024) as well as during visual working memory (Albers et al., 2013; Harrison & Tong, 2009; Rademaker et al., 2019), even though both of these processes have been shown to be largely unaffected in aphantasia (Keogh et al., 2021; Pounder et al., 2018), suggesting that they might not require conscious mental imagery. These findings suggest that the EVC represents internally generated visual information irrespective of whether these signals are instantiated through volitional, conscious imagery or not.”
Comment 2: From line 108 to 125, especially from 115 onwards when discussing how imagery inhibits activity in irrelevant neuronal populations, I suggest providing more clarification. When referring to modulating activity, does it include spontaneous and/or sub-threshold pre-mental imaging activity? If so, how is relevant activity distinguished from irrelevant activity? And why does inhibiting irrelevant activity promote the emergence of mental images? The 'sharpening' process linked to attention (Bartsch et al., 2017) and expectation (Kok et al., 2012) modulates feedforward (suprathreshold stimulus-related activation). What modulates the feedback of imagery? Clarifying this point is essential.
Response: I would like to thank the reviewer for pointing out this omission of essential information. This refers to spontaneous activity, with the idea that the feedback connections ‘carve out’ the imagined stimulus by inhibiting irrelevant activity. I have clarified this in the manuscript as follows on page 3, line 122:
“This is similar to the ‘sharpening’ process assumed to underlie attention (Bartsch et al., 2017) and expectation (Kok et al., 2012). Instead of modulating bottom-up signals like during attention and expectation, imagery is assumed to modulate spontaneous, baseline fluctuations in brain activity; carving out stimulus representations by inhibiting irrelevant activity (Koenig-Robert & Pearson, 2021; Pearson, 2019). Importantly, if imagery is instantiated through inhibition of irrelevant activity, overall activity levels in EVC might not increase and might even decrease compared to baseline (Fig. 1B).”
Comment 3: In the same section discussing the involvement of the visual cortex in imagery, what is the relationship between feedback (imagery) and retinal feedforward? Are they distinct neural populations in the early visual cortex, or are they the same population? Is there competition for resources between the two? Could there be temporal alternation? These questions are relevant and should be at least partially addressed in the text. The following excerpt illustrates my point, but the author may skip it if deemed irrelevant: studies on mice show that the primary visual cortex is involved in both feedforward and feedback processes at distinct time intervals (Keller et al., 2020). Wilming et al. (2020) found a dissociation between endogenous (feedback) and exogenous (feedforward) components in the early visual cortex (including V1) in terms of frequency bands. Gamma conveyed stimulus-dependent information, while alpha carried endogenous information related to perceptual choice. These findings suggest a temporal alternation between feedback and feedforward processes, possibly involving overlapping neural populations but at different times (Semedo et al., 2022, Contemori et al., 2022).
Response: This is an important issue that I agree needs to be discussed in more detail. It is related to the point in the discussion about how the brain keeps apart real (feedforward) and imagined (feedback signals. I would like to thank the reviewer for these references and I have now included them in this section on page 6, line 270:
“…Finally, other proposals have suggested that feedforward and feedback signals might be separated in time, where feedforward signals might be carried by faster oscillations in the gamma band while feedback signals are carried by slower oscillations in the alpha/beta band (Bastos et al., 2012, 2014; Keller et al., 2020; Semedo et al., 2022; Wilming et al., 2020).”
Comment 4: In the introduction, the early visual cortex is described as a retinotopic structure. Please note that the foveal cortex has been shown to be involved in non-retinotopic activity (i.e., Foveal feedback, Williams et al., 2008, but see Oletto et al., 2022, for a review). This non-retinotopic feedback process appears to be related to some form of imagery process involving mental manipulation of the stimulus, as its timing can be modulated by a mental rotation task (Fan, 2016). The author mentions at line 169 and onwards that the EVC represents fine-grained visual details during imagery, and without vivid details, it is not involved in imagery. This claim aligns with studies on non-retinotopic Foveal Feedback. For instance, Fan (2016) demonstrates that the feedback does not activate for blurred objects. Additionally, Contemori (2023) shows that interfering with feedback not only affects task sensitivity but also introduces perceptual biases in decision criteria. The line of research on Foveal Feedback seems closely connected to what is described in the review, and there are interesting links that could provide relevant insights into the topic.
Response: I would like to thank the reviewer for pointing me towards this interesting literature that I was unaware of. I will definitely read into it for future projects. However, the focus of this manuscript is limited to the early visual cortex because of its historic importance in the mental imagery debate and I have therefore decided not to include these studies in this specific paper.
Reviewer 3 Report
Comments and Suggestions for Authors
It was a pleasure to read this well-written article that focuses on a specific question in a broad field, i.e. the role of early visual cortex (EVC) in visual imagery. The article is well-embedded in the relevant literature, inspiring and of interest for the general readership of "Vision". I have to two suggestions:
(1) EVC and low-level features: The author makes the point that EVC-contributions depend on the nature of imagery and proposes that specifically imagery of low-level features would involve EVC. While most of the manuscript is moderate with this claim, at times it appears that imagery of low-level features is the exclusive criterion for EVC involvement. I presume the author does not want to be as exclusive and might need to adapt the wording accordingly. E.g., we know that EVC-responses can be modified from feed-back of higher areas (as described by the author), which thus impinge higher level properties onto EVC such that in turn not only low-level features become evident in EVC. Of course, this issue also depends on the definition of low-level features, which would need to be spelled out explicitly in this review.
(2) In "Conclusion" the differential interpretation of superficial vs deep layer involvement is described. Given the importance of the middle layers, as they comprise the input layer, they diserve attention here as well.
Author Response
Reviewer 3: It was a pleasure to read this well-written article that focuses on a specific question in a broad field, i.e. the role of early visual cortex (EVC) in visual imagery. The article is well-embedded in the relevant literature, inspiring and of interest for the general readership of "Vision". I have to two suggestions:
Response: I would like to thank the reviewer for their overall positive evaluation and helpful suggestions. I address each of them in detail below.
Comment 1: EVC and low-level features: The author makes the point that EVC-contributions depend on the nature of imagery and proposes that specifically imagery of low-level features would involve EVC. While most of the manuscript is moderate with this claim, at times it appears that imagery of low-level features is the exclusive criterion for EVC involvement. I presume the author does not want to be as exclusive and might need to adapt the wording accordingly. E.g., we know that EVC-responses can be modified from feed-back of higher areas (as described by the author), which thus impinge higher level properties onto EVC such that in turn not only low-level features become evident in EVC. Of course, this issue also depends on the definition of low-level features, which would need to be spelled out explicitly in this review.
Response: I would like to thank the reviewer for pointing out this important nuance. I agree that it is important to discuss that EVC activity is also influenced by high-level features. My main claim in the current paper is that EVC activity is especially sensitive to changes in low-level visual details which are exactly the features of imagery that are difficult to control, leading to difficulties in observing consistent EVC involvement in imagery studies. I have added this nuanace in the manuscript as follows:
I define low-level features on page 2, line 57:
“Together, this means that activity in the EVC is highly sensitive to changes in low-level features of visual signals, such as their exact distribution of edges, location and orientation (Hubel & Wiesel, 1968).”
Page 2, line 62:
“…irrespective of the specific low-level instantiation of that stimulus (Thorpe & Fabre-Thorpe, 2001). It is important to note that this distinction between low and high-level sensitivity is not discrete. Specifically, activity in EVC has also been shown to be modulated by high-level properties such as semantic scene context (Morgan et al., 2019; Muckli et al., 2015) and activity in high-level cortex has also been shown to be influenced by certain low-level features such as location (Kravitz et al., 2010). However, in general, activity in EVC is much more sensitive to variation in low-level features (Park & Serences, 2022). This means that, for example, the same neurons in high-level visual cortex might fire in response to a chair from many different viewpoints whereas the pattern of activity in EVC will be widely different in each specific instance.”
And I nuanced the following sentence on page 5, line 195:
“This means that EVC activity should is likely not always be observed during mental imagery, but should only be there mostly when low-level details are indeed imagined.”
Comment 2: In "Conclusion" the differential interpretation of superficial vs deep layer involvement is described. Given the importance of the middle layers, as they comprise the input layer, they diserve attention here as well.
Response: I agree with the reviewer that it is important to mention the middle layer in this discussion and have added it to the conclusion as follows. Page 6, line 268:
“Finally, given that imagery seems to be able to recruit the EVC in a similar way to perception, one important open question is how the brain keeps track of which visual signals reflect external reality and which reflect internal imagination. One hypothesis has been that the distinction is made based on whether the middle layer of EVC is involved, which is where bottom-up sensory signals arrive in the cortex (Lawrence et al., 2018). Another hypothesis is that the feeling of ‘realness’ is determined by whether or not the superficial layers of the EVC are activated (Bergmann et al., 2024). Initial results show that during mental imagery, only deep layers of EVC are involved. In contrast, superficial layers do get activated during seemingly ‘real’ illusory perception (Bergmann et al., 2024).”